# Imaging in Hip Arthroplasty Management—Part 1: Templating: Past, Present and Future

**DOI:** 10.3390/jcm11185465

**Published:** 2022-09-16

**Authors:** Edouard Germain, Charles Lombard, Fatma Boubaker, Mathias Louis, Alain Blum, Pedro Augusto Gondim-Teixeira, Romain Gillet

**Affiliations:** Guilloz Imaging Department, Central Hospital, University Hospital Center of Nancy, 29 Avenue du Maréchal de Lattre de Tassigny, 54035 Nancy, France

**Keywords:** hip, arthroplasty, CT, radiographs, 3D parameters

## Abstract

Hip arthroplasty is a frequently used procedure with high success rates. Its main indications are primary or secondary advanced osteoarthritis, due to acute fracture, osteonecrosis of the femoral head, and hip dysplasia. The goals of HA are to reduce pain and restore normal hip biomechanics, allowing a return to the patient’s normal activities. To reach those goals, the size of implants must suit, and their positioning must meet, quality criteria, which can be determined by preoperative imaging. Moreover, mechanical complications can be influenced by implant size and position, and could be avoided by precise preoperative templating. Templating used to rely on standard radiographs, but recently the use of EOS^®^ imaging and CT has been growing, given the 3D approach provided by these methods. However, there is no consensus on the optimal imaging work-up, which may have an impact on the outcomes of the procedure. This article reviews the current principles of templating, the various imaging techniques used for it, as well as their advantages and drawbacks, and their expected results.

## 1. Introduction

Hip arthroplasty (HA) is a frequently used procedure with high success rates. Its main indications are primary or secondary advanced osteoarthritis [1]. The goals of HA are to reduce pain and restore normal hip biomechanics, allowing a return to the patient’s normal activities [2]. To reach those goals, the size of implants must suit, and their positioning must meet, quality criteria, such as maintenance of leg length and femoral offset (FO), anteversion and inclination of the cup, and antetorsion of the femoral stem [2]. Those factors can be determined by preoperative imaging, and when they are not respected, patient dissatisfaction and mechanical complications can occur. Moreover, accurate pre-operative implant sizing could reduce surgical time and inventory needs [2]. HA preoperative planning used to rely on standard radiographs, suffering from magnification factor-induced errors, but recently the use of EOS^®^ imaging and CT is growing, given the 3D approach provided by these methods [2,3,4,5,6,7,8,9,10]. Especially, 3D printing offers an improvement in understanding patient-specific anatomy, thus enhancing patient outcomes (operation time, intra-operative blood loss, X-ray frequency, post-operative drainage), particularly in complex cases and for less experienced surgeons [11,12,13].

We present a narrative review of preoperative templating in HA, focusing on its classical aspects and emerging techniques, with their own advantages and drawbacks, driven by the hypothesis that those techniques have the potential to reinforce radiographic templating, but not yet to replace it.

## 2. Preoperative Planning

### 2.1. Background

Digital preoperative planning enables the surgeon to determine which prosthesis size to use and where to place it within the bone, aiming at optimal postsurgical functional restoration [14]. As oversized prostheses can increase the risk of periprosthetic fractures, and undersized prostheses can lead to dislocation, instability, and loosening, reliable prosthetic sizing and positioning is mandatory [5]. Thus, the aim of templating is to restore normal anatomy, especially the hip rotation center, femoral lateralization, and inferior limb length. Inadequate restoration of hip biomechanics is known to be a frequent cause of patient dissatisfaction. For example, 15–30% of patients complain of limb length discrepancy and up to 60% of THAs result in inadequate femoral offset (FO) [15]. Templating was classically performed on conventional radiographs, which are 2D and have a magnification factor that needs to be compensated. In this context, EOS^®^ and CT imaging represent valid alternatives with good reproducibility [16].

### 2.2. Hip Deformities

#### 2.2.1. Measurements and Their Implications

According to some authors and until recent years, surgery goals were to deepen the acetabular socket by reducing the AO, making room for the acetabular component, and increasing the FO to keep the GO constant [9,16]. Currently, the literature is not conclusive that the cup should be medialized and the femoral offset increased in compensation. In practice, cup placement should rather be anatomical [17]. A fully medialized reconstruction cannot be justified for now, especially in young patients who need to conserve their bone stock in case of revision surgery, as only small biomechanics gains were noted in a whole-body simulation of a gait cycle by De Pieri et al. [18]. Moreover, the acetabular offset should not be reduced, as it requires the use of stems with greater offsets than the natural femoral one, which induces a risk of impingement [17]. When severe hip joint deformity is present, the contralateral hip can be used for templating [19], but no clear recommendations exist in the case of bilateral severe deformity. Amongst preoperative measurements, some can also be used to evaluate postoperative outcomes in HA, among which are lower limb length, global offset (GO) (sum of acetabular (AO) and FO), and femoral neck antetorsion (FNA). These parameters can be measured on pelvic AP radiographs or EOS^®^ imaging, and on coronal CT-reformatted images in the plane created by the femoral neck axis and the femoral mid-shaft axis [15]. Femoral offset measurement is more precise using 3D CT-scan reconstruction as it does not depend on test conditions, because the frame is placed within the femoral axis and is not influenced by position inconsistencies or hip pathologies [20].

AO corresponds to the distance between the femoral head center and the acetabular floor or the midline, FO to the distance between the femoral head center and the femoral mid-shaft axis. The cervico-diaphyseal angle (CDA) should also be reported (Figure 1).

FNA measurements have traditionally been done on radiographs but are not sufficiently precise for templating. EOS^®^ imaging or CT-based measurements are therefore recommended (Figure 2) [16].

Hip function depends on the proper orientation of the muscles about the center of rotation of the joint (i.e., limb length imbalance and offset correction) [16]. An imbalance in GO may lead to limping due to abductor dysfunction. Reduced FO can lead to acetabular-polyethylene wear, dislocation, and loosening [15], whereas an increase can lead to residual pain and limping by exacerbation of muscle and soft-tissue tension. However, it is still unclear which of these parameters (GO, FO, or AO) should be taken into account for offset correction. FNA below 10° is supposed to be deleterious to the long-term outcome for cemented stems [16].

Much less attention has been given to the sagittal orientation of the stem. With the recently increased use of the anterior surgical approach, and the rise in popularity of the femoral short stems, the femoral component might get implanted in a flexed position. Yoshitani et al. did not find significant differences concerning radiological and clinical follow-up at approximately 5 years between flexion and neutral alignment, but long-term studies are required [22]. This measurement is difficult to assess precisely on lateral radiographs [22]. As for FNA, CT represents a valuable tool (Figure 3). EOS^®^ must be evaluated in this setting. An anterior acetabular offset can be measured (distance between the center of the femoral head and the retrocondylar axis in the axial plane), but its usefulness remains to be determined.

#### 2.2.2. Types of Architectural Deformities and Their Implications

A recent classification has been proposed by Kase et al. to distinguish different types of architectural hip deformities and their associated abnormalities, according to femoral head-translation patterns, using the distance between the femoral head center and the acetabular center [15]. In this study, femoral heads were centered in 61% of the patients studied. In 26% of patients, hips were lateralized and often presented femoro-acetabular osteophytes. There were only 4% of medialized hips (often accompanied by acetabular protrusion), 3% of proximally displaced hips (flattening of the femoral head or acetabular deformation), and 6% of proximo-lateral displacement (femoral head and acetabular superolateral deformity) (Figure 4).

Proximal and proximo-lateralized displacement induced considerable limb shortening. Femoral head translation on radiographs or CT has to be taken into account to correct limb shortening and/or pathologic offset [15], as it allows surgeons to restore preoperative anatomy. More precisely, in centered hips, AO was similar to healthy hips [24], and reproducing native anatomy could restore the hip rotation center [15]. On the other hand, in medialized hips, one should consider rotation center lateralization to avoid impingement and FO reduction to avoid excessive soft-tissue tension with a potential clinical impact [15].

### 2.3. Principles and Methods

#### 2.3.1. Radiographs Technical Aspects

Templating has conventionally been made by applying tracing papers on radiographs but has become impractical with the implantation of digital radiographs, which presents a similar reproducibility [25]. The different measurements necessary for hip templating are summarized in Table 1.

In the authors’ institution, radiographs include an anteroposterior pelvic view, from the iliac crest to the femoral proximal third, to visualize the femoral stem destination. The patient should be standing with 15° of internal hip rotation if he can, whereas, in case of fracture, he might be lying down on his back. The X-ray beam is centered 2 cm under the pubic symphysis and its source at 1.50 m, which corresponds to a magnification factor of 1.15, like the tracing papers supplied by the manufacturers. The magnification factor might be variable amongst the distance X-ray source/detector, and femoro-acetabular joint/detector; therefore, it is dependent on the patient body habitus and position variations induced by pain, potentially leading to limitations in radiographic templating [26].

While standing radiographs are obtained without difficulties in most ambulatory patients, it might be impossible in emergency situations or in case of advanced disease. This issue is largely compensated by the fact that it is reasonable to consider the pelvic position in the supine position, as the acetabular component had an optimal orientation in 90% of the cases in the study by Nishihara et al. [27]. Those results were recently confirmed by Uemura et al., since the pelvic sagittal inclination changes from supine to standing were smaller than 10° in approximately 80% of the cases in their study [28], and pelvic positions in supine and standing postures were reproducible in a second recent study by the same team [29]. The pelvic position in the supine position at 10 years of follow-up was shown to be a good functional reference by Tamura et al., as it did not show variation over time, unlike the standing pelvic sagittal inclination [30]. On the other hand, in some patients with developmental dysplasia of the hip, the acetabular version differed between the supine and standing positions, so that Tani et al. recommended the use of the values obtained in the standing position for preoperative planning [31], and Tachibana et al. [32] and Bhanushali et al. [33] recommended assessing postural changes (radiographs in both standing and supine positions) in the sagittal pelvic tilt in case of dysplasia and called for other studies to determine how those postural changes affect the biomechanical environment of the acetabulum. Therefore, in the authors’ opinion, even though radiographs have tended in the last years to be realized in the standing position to reproduce functional position, supine radiographs can be obtained in any patient with good confidence in pelvic position, especially in the emergency setting. However, both standing and supine radiographs should be obtained, if possible, in patients with hip dysplasia.

With radiographs, the first step is to determine the magnification factor by using an existing body implant or a radiographic marker of known dimensions [14].

Then, the pelvic axis must be determined, most commonly by drawing a line between the iliopubic branch contours on anteroposterior pelvic radiographs (e.g., teardrop sign) (Figure 5). On radiographs, limb length discrepancy can be calculated by drawing a line perpendicular from the inter-teardrop axis to the top of the lesser trochanter. One should consider 5 mm as a cut-off, as symptoms are infrequent beyond this value [34,35,36].

If available, an EOS^®^ acquisition can be performed and the inter-teardrop axis will also be used to calculate the appropriate inclination angle of the acetabular component [14].

##### Acetabular Cup Templating

The femoro-acetabular joint rotation center can be determined by placing a digital acetabular template at an angle of approximately 40–45° to the pelvic axis [14], or by applying different tracing papers to match acetabular morphology, so that the acetabular component can be positioned with an inclination of 40–50° and an anteversion of 20–30° to avoid bony impingement [37]. Concerning hip stability, the “acetabular safe zone” initially described by Lewinneck et al.—of 40 ± 10 degrees and 15 ± 10 degrees, respectively [38]—has been refuted and is more likely to be multifactorial and patient-specific, so that new unique values have not yet been clearly defined [39,40] and the hypothesis of a static safe zone is simplistic [41]. More precisely, the optimal positioning of the acetabular cup is thought to depend on sagittal pelvic mobility. Some authors, therefore, call for the development of standardized algorithms for the placement of kinematically aligned acetabular components [42].

However, important variations between operative and radiographic angles have been reported [17]. Cup inclination can be visualized on an AP pelvic view or with an EOS^®^ imaging system. Cup anteversion measurement is more variable. CT with multi-planar reformation has been shown to be more accurate than intraoperative measurements or radiographs [43]. EOS^®^ imaging can also be used. Several methods exist using radiographs. The most common method used to be the one proposed by Woo and Morrey [44], corresponding to the angle formed by a line drawn tangential to the face of the acetabulum, and a line perpendicular to the horizontal plane, as seen on a lateral view of the pelvis. However, recently Lee et al. demonstrated that using AP radiographs (pelvic or hip), the methods by Pradhan et al. [45], Liaw et al. [46], and Lewinnek et al. [38] might provide accurate anteversion measurements with high reliability, regardless of the type of radiographs (hip or pelvis) [47]. All those imaging data do not seem sufficient, as Grammatopoulos recommended implanting the cup in 5° less inclination and 8° more anteversion than planned to achieve the target radiographic position [48]. To guide angular positioning, a jig or a proctator placed on the inserter handle can be used. Presently, patient-specific measurements are being developed, and classical approaches are clearly called into question.

##### Femoral Stem Templating

The size of the femoral component is determined by measuring the width of the endosteal canal distally within the femoral diaphysis and metaphysis; its position is chosen by placing it within the femur in a position to reproduce limb length or correct any discrepancy, obtain an appropriate FO, and match the center of the femoral head with the center of rotation of the joint. This positioning must consider the length of femoral neck resection (measured proximal to the lesser trochanter or distal to the greater trochanter), which can vary to achieve postoperative goals, as the prosthetic neck length, cervico-diaphyseal angle in the case of a modular implant, and height and diameter of the femoral head can be adjusted if needed. Osteointegration depends on several factors, including bone quality. The Dorr classification [49] aims to guide indications for the type of femoral component fixation (e.g., cemented or uncemented) and evaluates the risk of perioperative fracture of the proximal femur. It is based on the cortical index, corresponding to the ratio of the canal diameter, 10 cm distal to the midportion of the lesser trochanter divided by the inner canal diameter at the midportion of the lesser trochanter (Figure 6), and the femoral cortical aspects on radiographs (AP and lateral views) [50,51].

The Singh index analyzes the trabecular pattern of the proximal femur, classifies osteoporosis into six grades (grade 6 represents normal bone density and grade 1 reflects severe osteoporosis), and is available for routine use and mass screening [52]. It has been shown that Dorr types were correlated with occult osteoporosis in postmenopausal women with osteoarthritis, and these radiographic features have been postulated to be determinants of fracture risk and prosthesis longevity [53]. However, a clearly defined role of the Singh classification has not yet been described for HA templating.

### 2.4. Perspectives

#### 2.4.1. CT-Scan

##### Background

Even though radiographic templating is a well-known process in HA planning, it is also admitted that surgeons need better methods due to the magnification factor and patients’ position differences using standard radiographs [54]. CT has gradually gained interest and will probably complement radiographs in templating as it is more accurate in planning implant size, component alignment, and postoperative leg length than radiographs. Additionally, with recent advances in CT technology and image reconstruction algorithms—including deep-learning reconstruction—radiation dose exposure can be potentially reduced to values similar to radiographs [55]. CT can also provide 3D information and assist surgeons intraoperatively [7,8,9,16,55,56]. Furthermore, it allows a precise evaluation of bone stock and of the osteophytes that should be resected. Inoue et al. stated that CT-based 3D templating made it possible to achieve reproducible stem antetorsion (between pre- and postoperative CT scans) and choose accurate stem and cup sizes in the case of developmental dysplasia of the hip [6]. Madadi et al. underscore that if inclination and anteversion were crucial for acetabular cup placement, which can be performed considering four directions (e.g., inward, outward, upward, and downward), osteophytes might displace the femoral head and acetabular fossa. These latter were well depicted on CT (e.g., central osteophytes and hypertrophic OA) and their preoperative depiction was deemed crucial for planning [57]. For the diagnosis of acetabular bone loss in revision surgery, an expert panel stated that an AP pelvis radiograph is sufficient only in the case of minimal bone loss. CT should be considered in more severe cases, especially in case of fracture, concomitant rotation of the hemipelvis, extensive osteolysis, and medial migration of the acetabular component [58].

Finally, one should be aware that CT for preoperative planning rather than radiographs might lead to the discovery of incidental findings (e.g., acute diverticulitis, masses, osseous tumors, aneurysms, or abdominal wall hernias), which lead to delay or cancelation of arthroplasty in 5% [59].

CT could therefore be systematically performed preoperatively for standard measurements, especially in the case of the presence of osteophytes and for revision surgery, and if specific low-dose reconstruction algorithms are available. CT-based navigation systems and 3D-printed templates still need more studies to be systematically recommended as they might be costly and time-consuming.

##### Technical Aspect

CT scan measurements have been defined in Figure 1 and Figure 2.

In practice:(1)the hip can be classified as mentioned above (e.g., centered, medialized, lateralized, proximalized, or proximo-lateralized);(2)the pre-arthritic centers of the femoral head and acetabulum must be determined (potentially using the contralateral hip if healthy), and the optimal diameter of the acetabular cup measured on a transverse CT slice, so that its template can be positioned at the level of the true acetabular floor medially and of the subchondral bone proximally, slightly superior and medial to the center of the native acetabulum to simulate reaming;(3)the stem size and model can be determined, such that the templated head center can match the templated cup center craniocaudally, with the native mediolateral center maintained original even if pathologic, except in case of medialized head, which has to match templated cup center in both axis [60].

CT-based preoperative planning usually relies on 3D templating software, but Chinzei et al. demonstrated that templating using CT multiplanar reconstructions alone is more available and may be useful as a complementary tool without additional costs [4]. Practically speaking, a multiplanar reconstruction can be constructed without additive software, as most picture archiving & communication systems (PACS) include 3D reformat options. First, the femoral shaft axis has to be determined in the frontal and sagittal planes, then this image is reproduced on the plane passing through both the axis and the center of the femoral head [4]. This technique also allowed the authors to measure femoral head cup diameter, stem size, length of the modular neck, and distance from the neck osteotomy. Then, transparent template sheets were applied to the screen and the image externally rotated until the lesser trochanter was displayed, to finally determine the neck osteotomy level [4]. However, this procedure was time-consuming and does not seem practical, in our opinion. Moreover, its inter- and intraobserver correlation were not assessed.

Geijer et al., using a 3D templating software and low-dose CT with an acquisition from hip to knee, showed near-perfect inter- and intraobserver agreement in measuring AO, FO, and FNA. These authors stated that using 3D datasets practically eliminated the need for exact patient positioning, in pre- and postoperative CT scans [16]. One must keep in mind that there are various methods for FNA measurement described in the literature hampering the comparison of different studies [16]. Regardless, CT scan measurements of femoral antetorsion remain the gold standard, as shown in Figure 2 [61]. Automated measurement algorithms might be available in the near future, as Veilleux et al. showed an effective automated technique for determining pelvic and acetabular orientation, using 3D images from CT scans, as an aid in preoperative planning, which is therefore less time-consuming than manual calculation [62].

CT-based templating has improved imaging measurements and postoperative results in terms of component placement. Nishihara et al. stated that the use of a CT-based navigation system improved cup positioning compared to freehand cup placement, even in the supine position with a direct anterior approach, though it is thought to be as reliable as CT-based navigation [8]. Scheerlinck et al. even declared that based on CT-3D templating in non-deformed femora, the non-modular femoral stem could restore the anatomical hip rotation center so that failure to restore anatomy might be due to surgical inaccuracy rather than lack of implants matching the patient’s native anatomy [9]. More than classical imaging measurements and standardized values, 3D data obtained from CT have been shown effectively in intraoperative simulations using 3D-printed materials [7,56]. Those advantages are also advocated in revision surgery in the case of acetabular bone loss, as CT allows a 360° assessment of bone loss with a pelvic 3D rotation, a better assessment of osteolysis, a segmentation to evaluate the pelvis with the implant subtracted, and the generation of 3D-printed materials which can help in implant choice and design [58].

Three-dimensional printing technology, also called rapid prototyping, allows to create 3D scale models of physical objects quickly, using imaging data and thermoadhesive materials such as thermoplastic or liquid metals. It is supposed to improve osteointegration in acetabular implants, developmental dysplasia of the hip, and generally help the surgeon in their operative planning, but it is not yet available for soft tissue preoperative imaging [63,64].

##### Main Strengths

In our opinion, the main strengths of CT are its ability to provide precise measurements and its 3D capacities, allowing the use of software and reducing the need for optimal patient positioning, which could improve patient workflow and reduce image post-treatment additional work time.

##### Limitations

3D CT templating remains complex, costly, and not widely available [60]. Its indication also remains controversial, as to whether it should be limited to special issues such as hip dysplasia or advanced degenerative changes, or recommended for all HA procedures [60]. Direct costs of a preoperative CT were reported to range from 53 to 116 euros in a German study, and thus considered low [65], but remained more expensive than radiographs. The radiation dose used to be of concern, but is nowadays decreasing, and tends to approach that of radiographs [16,60,66]. Kobayashi et al. combined the reliability of CT with the simplicity of acetate templating, by applying templates on real printed CT images in adjusted coronal and axial planes (parallel to the neck axis and to the femoral shaft axis), including the lesser trochanter in the frontal plane and the femoral head maximal diameter in the axial plane. Even though they recognized CT advantages, they could not demonstrate the superiority of CT scans over radiographs. Therefore, they stated that high-quality radiographs are sufficient for now until reliable surgical tools and post-operative image acquisition become available and affordable in the surgeon’s routine using CT but mentioned that they would continue CT-based templating as it remains more accurate in their opinion [60].

#### 2.4.2. EOS^®^ Imaging

##### Background

It has recently been shown that planning software, based on radiographs obtained with the EOS^®^ imaging system (EOS^®^ imaging, Paris, France), could also be useful in templating, as it was more accurate than 2D radiographs and equal to CT [2]. It corresponds to a low dose biplanar digital radiographic imaging system involving gaseous photon detectors, used in over 400 medical centers worldwide. It takes approximately 20 s for an adult full-body scan [67], therefore a little bit longer than CT, and quite shorter than radiographs, as several views are acquired one at a time. Contrary to CT and radiographs, the EOS^®^ imaging system allows the simultaneous acquisition of two orthogonal radiographic images without magnification factor, and considers lower limb deformities, with a patient in a functional standing or sitting position inside the system [68], allowing to study the variation of the sagittal acetabular version. This system then creates a 3D reconstruction for parameter calculation [68], so that it is used as a gold standard in limb length discrepancy compared to radiographs [69]. The same study can provide femoral length, tibial length, and hip, pelvic, and spine parameters (beyond the scope of this paper) (Figure 7 and Figure 8). Mayr et al. showed a strong overall correlation between the EOS^®^ imaging system and CT scan measurements, and high inter- and intrareader reliability in measuring the femoral antetorsion angle, but in case of torsional malalignment, EOS^®^ did not correlate with CT, and presented an advantage as it does not depend on legs’ positioning [61]. It is also proposed to be used in the postoperative follow-up, as measurements are relatively quickly realized [67], coupled with the low-dose advantage [2,26] and the absence of metallic artifacts from implants [70]. Some authors have even proposed to replace standard radiographs with the EOS^®^ imaging system [2], in the pre- and postoperative setting, but this attitude requires more studies to be sustained, for as another study points out, difficulties exist in defining reference points on the 3D images provided by EOS^®^ imaging after THA [71]. Of note, this technique can also be used in limb length discrepancy measurement in children, knee architectural disorders, and spine disorders analysis. It has recently been used to show altered hip functional outcomes postoperatively when femoral malrotation occurred after femoral shaft intramedullary nailing in patients with fractures [72].

Concerning radiation dose, it is 4–30 times lower than that of CT [73] and 6–9 times lower than that of radiographs [74]. In the United States, its cost is about ¼ to 1/6 relative to CT [68].

In our opinion, this technique could be performed in conjunction with hip radiographs or CT to study the spinopelvic complex, the spine, and the lower limbs, and to consider the hip in its whole environment, especially in the case of lower back pain or prior surgery.

##### Practical Aspect

Practically speaking, preoperative images are modeled in 3D using sterEOS^®^ software, in which different points are manually positioned to obtain automatic data-sets [61], and the hipEOS^®^ planning software (EOS^®^ imaging, Paris, France) integrates the manufacturers’ 3D component templates into the modeled bones [2]. Using this software, surgical planning can be performed by the surgeon, with the determination of the size of the femoral stem and of the acetabular cup, the change in leg length, and FO [2]. For primary THA, Knafo et al. showed that, in conjunction with a navigation system, the EOS^®^-based planned acetabular and femoral component size corresponded with that implanted ± 1 size in 100% and 94% of the cases, respectively. They also found a postoperative leg length of 1.9 ± 5.9 mm compared to the planned value, and an agreement between the postoperative and the planned FO value of 0.3 mm (SD ± 5.6), which was also acceptable and inferior to that of CT [2,66].

Huang et al. also described better performances of EOS^®^ preoperative planning compared to radiographs, using a digital-templating system without any software [5].

##### Main Strengths

In our opinion, this technique should be increasingly used as it provides 3D information based on 2D acquisition, also to be used with software, at a price lower than that of CT and similar to that of radiographs, with a dramatically decreased radiation dose. It also allows a potential whole-body acquisition on a single occasion, highlighting a combined approach of lower limbs, hips, and spinopelvic complex, which may be key in acetabular cup templating in the near future, especially with the possibility of combined imaging in the standing and sitting position.

##### Limitations

The soft-tissue analysis is impossible with an EOS^®^ imaging system. EOS^®^-based templating has not yet been investigated in the case of hip dysplasia, previous acetabular surgery, and revision surgeries. Its use is limited in the case of previous hip or knee prosthesis, as the sterEOS^®^ software cannot be used in the presence of implants [2]. It also features limitations in patients who cannot stand and is susceptible to movement artifacts [61]. In those cases, CT remains a valuable alternative.

##### Rationale for Preoperative Planning

Even though surgeons are responsible for implant choice and preoperative templating, radiologists must be aware of the different measurements required and not solely rely on the manufacturer’s software’s automatic measurement. Radiologists are guarantors of the quality and reliability of imaging studies, which must provide adequate support for templating without increasing radiation dose. Even though radiographic templating remains the reference, its limits are well-known and can be reduced with the use of CT and EOS^®^ imaging, which are both thought to be more precise without increasing radiation dose [2,5,61,67,68,69,70,71,72,73,75]. Strengths and limitations of 2D and 3D techniques are summarized in Table 2. Therefore, to improve functional results and avoid HA complications, radiologists must be aware of the advantages and limitations of the measurement methods of each imaging technique, keeping in mind that even using CT, which is thought to be the most reliable, templating should still be used as a guide rather than an absolute model [60]. For example, the preoperatively measured and planned stem orientation was never achieved in Belzunce’s study with CT (discrepancy of −1.4 ±  8.2 degrees with a 95% confidence interval of [−16.9, 13.8]) [76], and the preoperatively determined femoral stem and acetabular component are reliable in almost all cases, but within one size with EOS imaging, for example [2]. Moreover, Cech et al. stated that implanting components of different sizes than planned did not compromise THA outcomes, whereas medialized hips had worse outcomes, therefore underscoring the need of considering the hip in its whole geometry rather than in terms of strict quantitative measurements [3]. Concerning shoulder arthroplasty, CT-based templating has become a standard procedure and is now well-known to improve patients’ outcomes [77,78,79,80]. Tridimensional data have even been shown to be more effective than 2D datasets [79,80]. In the same way, knee arthroplasty templating using 3D data is becoming more and more popular, derived from 2D radiographs [81] or CT [82], sometimes with printed templates [83] and even machine-learning contributions in implant choice [84]. In this context, it seems reasonable to believe that technological advances in imaging will also improve patients’ care and the technical aspects of HA. Regardless of the other methods used for templating (CT and/or EOS^®^), radiographs remain fundamental for HA preoperative templating for comparison purposes with a postoperative radiographic follow-up. In our opinion, for straightforward situations such as non-deformed centered hips without limb length discrepancy or lumbar spine pathology, conventional radiographs are generally sufficient, as this strategy might be cost-effective. Hip structural deformities require CT, including the femoral condyles in the acquisition to assess acetabular morphology and FNA [12,56], even though it remains costly, as EOS^®^ imaging has not yet been investigated in this context, and post-operative complications might induce many more additional costs than an optimal preoperative imaging work-up. 3D CT-based templating seems to be the most promising technique in implant choice, while standard CT is essentially useful in depicting hip anatomy and degenerative changes. If a CT-based navigation system is available, it might be considered for acetabular cup placement. In both scenarios, EOS^®^ imaging should be performed, if available, to acquire weight-bearing information concerning both the lower limbs and spinopelvic complex, and 3D templating using software must likewise be considered [2,5,68,70,73,75,85]. If EOS^®^ is not available, one must keep in mind that CT-based measurements tend to limit positional variations. Finally, in our opinion, to clearly define if those techniques should replace radiographs or not, their use for the postoperative follow-up should be investigated and compared to radiographs.

## 3. Conclusions

The HA preoperative imaging work-up classically relies on pelvic and hip radiographs, but two main evolutions seem to delineate, potentially improving postsurgical functional outcomes: the EOS^®^ imaging system, which evaluates the whole physiological environment of the hip, including the spinopelvic complex and lower limbs, and seems to replace radiographs in lower limb discrepancy; and CT, allowing optimal assessment of acetabular morphology, FNA, and femoral flexion/extension, and supporting dedicated software to propose patient-specific implant designs, especially with the recent development of 3D-printing, which has been shown to be helpful in complex cases and developmental dysplasia of the hip and to increase postoperative outcomes. To date, those procedures might not yet replace radiographs but should be considered as complementary imaging technique, to improve patients’ outcomes. Even though preoperative templating remains a guide and should not be considered ideal in any case, we believe that for non-deformed centered hips without excessive degenerative changes, lower limb, or lumbar spine pathology, conventional radiographs might remain sufficient, whereas deformed and frankly degenerative hips, revision surgery, cases with concomitant lumbar spine, or lower limb discrepancy should be more widely explored (CT and/or EOS^®^ imaging), as preventable complications might occur in case of insufficient imaging procedures in the preoperative work-up.

## Figures and Tables

**Figure 1 jcm-11-05465-f001:**
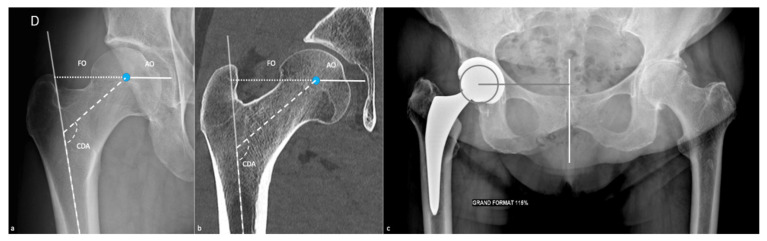
Preoperative measurements. Those values can be measured on (**a**) an anteroposterior pelvic radiograph and on (**b**) a coronal CT image, in the plane established by the femoral neck axis and the femoral midshaft. AO corresponds to the acetabular offset (white line: distance between the femoral head center (blue circle) and the acetabular floor), FO to the femoral offset (little-dotted line: distance between the femoral head center and the femoral midshaft axis (mild transparent white line)), CDA to the cervico-diaphyseal angle (large-dotted white line). A measurement of AO from the pelvic midline is shown on (**c**) an AP pelvic view [21], as it is more suitable in case of hip prosthesis, especially in case of cup protrusion.

**Figure 2 jcm-11-05465-f002:**
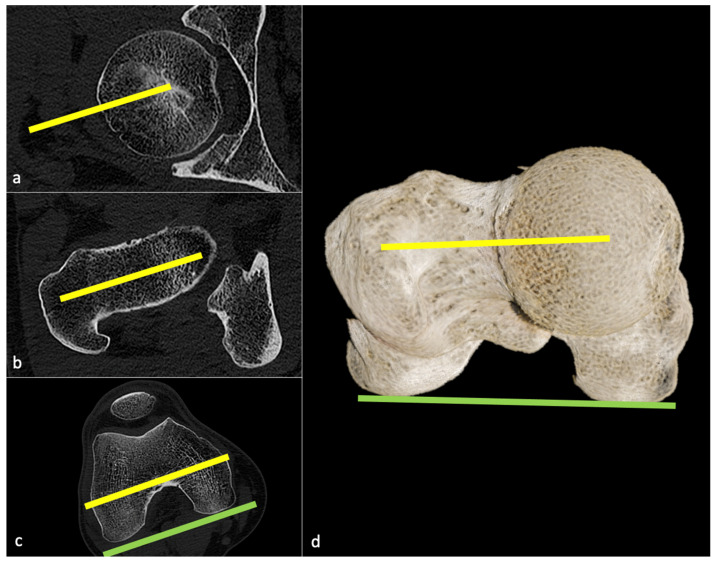
Femoral neck antetorsion measurement. Three axial CT slices must be selected: one shown in (**a**) at the femoral head center, one in (**b**) at the femoral neck to measure the femoral neck axis (yellow line), then one in (**c**) at the level of the roman arch to determine the intercondylar axis (green line). In (**d**), a global illumination reformat is shown to illustrate the 3D rendering of this measure, corresponding to the angle between the yellow and green lines.

**Figure 3 jcm-11-05465-f003:**
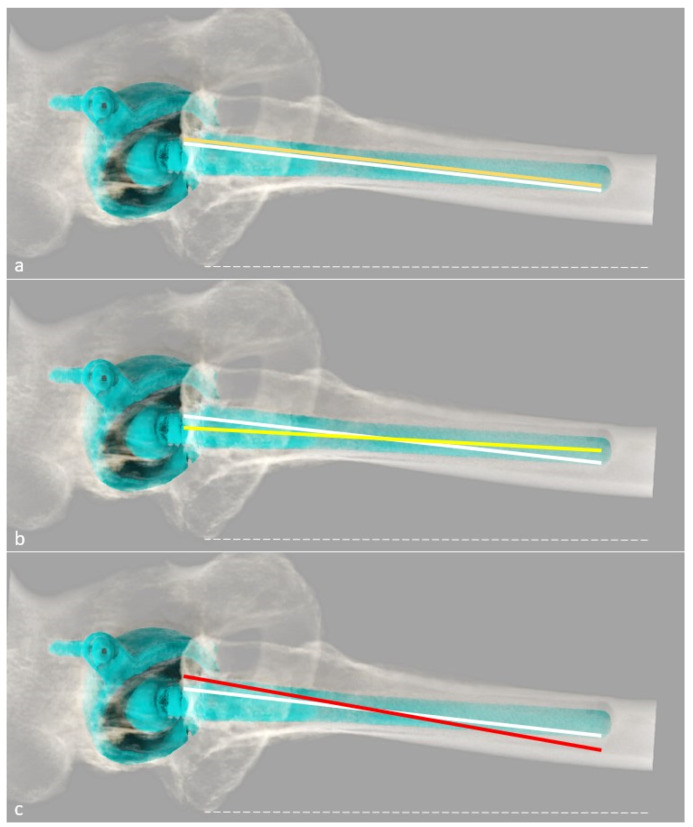
Representation of the flexion/extension of the femoral stem, using 3D CT-scan reformat, adapted from Abe et al. [23]. The dotted white line represents the retrocondylar axis, the white line the sagittal femoral tilt, and the colored line the stem axis. A theoretical neutral position is shown in (**a**) with the orange line; a negative value superior to −3° between the femoral tilt and the sagittal stem tilt is defined as flexion in (**b**) with the yellow line, which is the actual axis of this prosthesis; and a positive value superior to 3° is defined as an extension in (**c**) with the red line.

**Figure 4 jcm-11-05465-f004:**
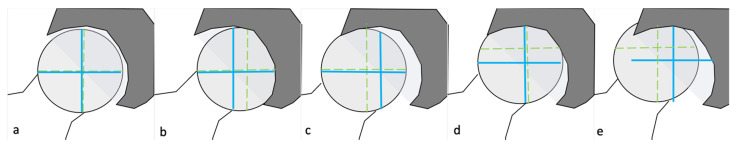
Hip deformities. Five types of hip deformities are shown, adapted from Kase et al. [11]. On each scheme, the femoral head is colored in grey, the acetabulum in dark grey, and the acetabular cavity in transparent grey. The blue lines correspond to the acetabular center and the green dotted line to the vertical and horizontal axis of the femoral head. In (**a**), a centered hip is shown as both axes are superimposed; in (**b**), a medialized (medialization of the vertical axis of the femoral head with respect to the acetabular one); in (**c**) a lateralized (lateralization of the vertical axis of the femoral head with respect to the acetabular one); in (**d**) a proximalized (cephalic displacement of the horizontal axis of the femoral head with respect to the acetabular one); and in (**e**) a proximo-lateralized (cephalic displacement of the horizontal axis of the femoral head with respect to the acetabular one, and lateral displacement of the vertical axis of the femoral head with respect to the acetabular one). An arbitrary cut-off of 3 mm was used by the authors to consider a displacement in each plane.

**Figure 5 jcm-11-05465-f005:**
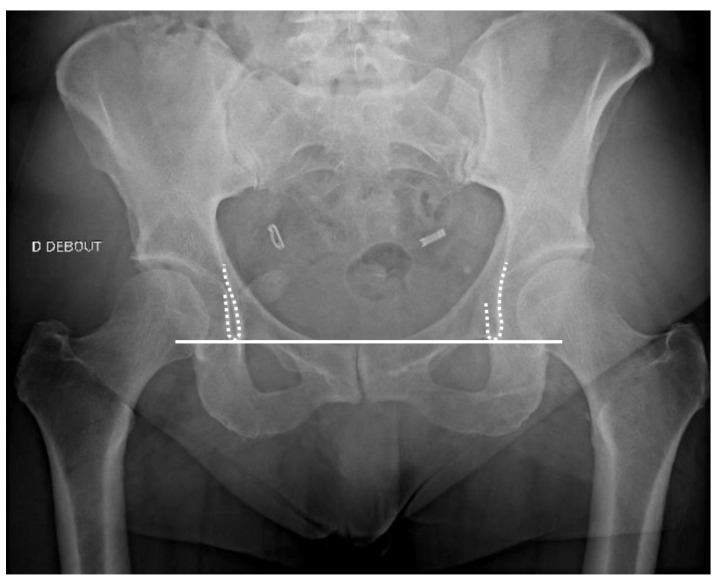
Inter-teardrop axis. The inter-teardrop axis shown on an anteroposterior pelvic radiograph (white line), the teardrops being represented by the dotted lines.

**Figure 6 jcm-11-05465-f006:**
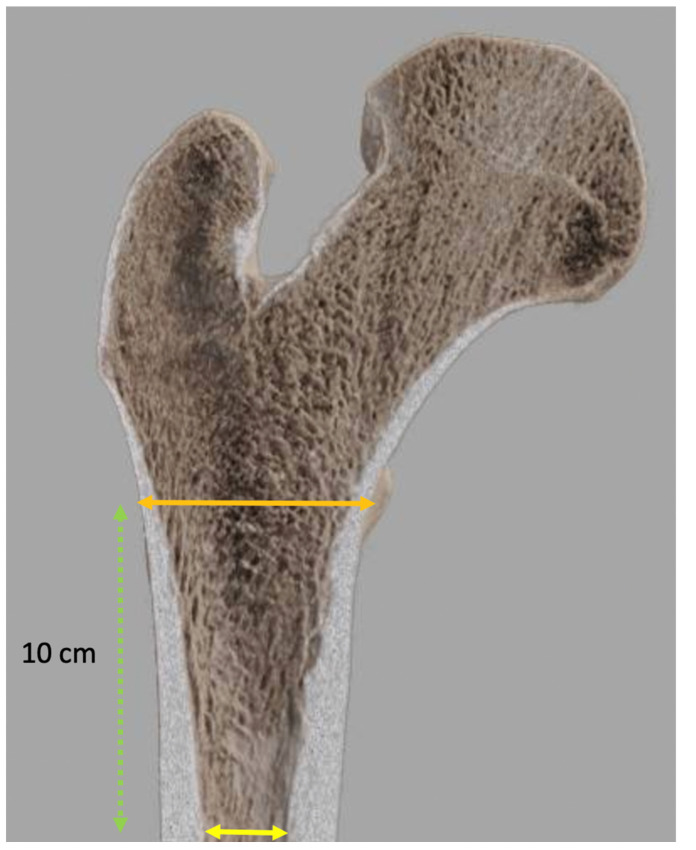
Cortical index calculation. Cortical index is calculated by measuring the ratio between the diaphyseal diameter between the cortices at the level (orange double-headed arrow) and the inner canal diaphyseal diameter 10 cm below the lesser trochanter (yellow double-headed arrow).

**Figure 7 jcm-11-05465-f007:**
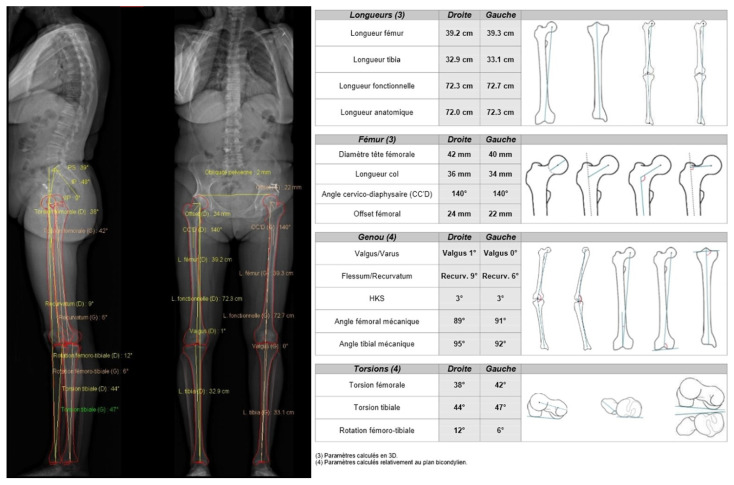
EOS^®^ imaging example showing pelvic and lower limbs measurements. In this example, pelvic parameters are shown (PS: sacral slope, IP: pelvic incidence, VP: pelvic version), and multiple lower limbs measurements are available (femoral and tibial length, femoral head diameter, femoral neck length, cervico-diaphyseal angle, femoral offset, femoral and tibial version/torsion, knee valgus/varus, hip-knee shaft angle, femoral flessum/recurvatum).

**Figure 8 jcm-11-05465-f008:**
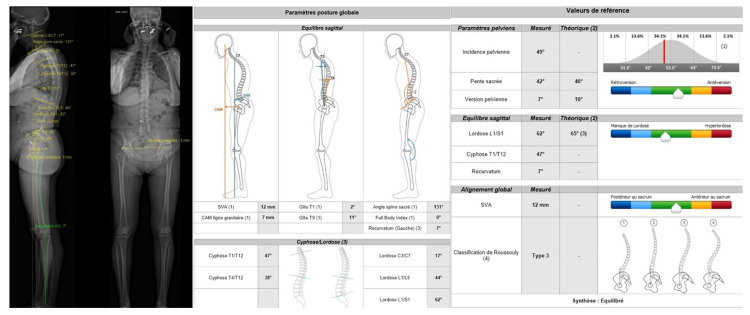
EOS^®^ imaging example showing spine parameters and their relationship with pelvic parameters. Spinal kyphosis and lordosis angles are provided along with the sagittal vertical axis measurement, as well as pelvic parameters, to provide a global appreciation of the spinopelvic complex, considered well balanced in this example.

**Table 1 jcm-11-05465-t001:** Measurement to be made on preoperative radiographs for hip arthroplasty templating.

Parameter	Utility
Magnification	Measurement adaptation
Pelvic axis (teardrop sign)	Limb length discrepancy
	Frontal inclination of the acetabular component
Femoro-acetabular joint rotation center	Acetabular component positioning and size determination
Width of the endosteal diaphyseal and metaphyseal canal diaphysis	Size of the femoral component
	Femoral offset determination
	Matching of femoral head and joint centers
	Positioning of the femoral component
Femoral neck resection length	Positioning of the femoral component
Cervico-diaphyseal angle	
Height and diameter of the femoral head	

**Table 2 jcm-11-05465-t002:** Strengths and limitations of 2D and 3D templating imaging techniques.

	Radiographs (2D)	CT and EOS Imaging (3D)
Advantages/Strenghts	- Reference technique - Commonly and widely used - Still considered essential for follow-up - Used for Dorr classification - No metal artifact	EOS imaging: - Considers spinopelvic complex mobility - More precise and reproductive than radiographic measurements (lower limb length+++) - Semi-automated - No metal artifact - Lowest radiation dose CT: - Osteophyte depiction (acetabulum) - Bone stock analysis, reconstruction with substraction of the implants, vascular analysis, for revision surgery +++ - More precise implants design and positioning - More precise for sagittal inclination of the stem, femoral offset and femoral neck antetorsion measurement - 3D printing: patient-specific design, effective intraoperative simulation - Not depending on patients’ position - Preoperative soft tissue (muscle and tendons) partial analysis
Drawbacks	- Magnification factor - Variation between operative and radiographic measurements - Femoral and limb length measurements depending on patient’s position and lower limb rotation - No soft tissue analysis	EOS imaging: - Less reproducible in the postoperative setting - No soft tissue analysis CT: - No acquisition in standing position - Radiation dose exposure (without low dose reconstruction algorithms) - Cost - Time consuming post-treatment - Metal artifacts

## Data Availability

Not applicable.

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
