# Peer review of "Imaging in Hip Arthroplasty Management—Part 1: Templating: Past, Present and Future"

_jcm, 2022, doi:10.3390/jcm11185465_

Round 1

Reviewer 1 Report (New Reviewer)

The authors review the state-or-the-art of the templating by comparing the standard radiograph, CT, and EOS-imaging technics. The manuscript is well written and presents interesting discussion. The following comments need to be addressed.

1.      CT and EOS-imaging provide 3D information of bones. The authors are recommended to more clearly discuss/summarize (such as a table which compares the difference) how that 3D information may improve the accuracy of the parameters obtained by standard radiographs, and how those improvements may be helpful in terms of reconstruction of normal hip biomechanics.

2.      Line 139 specified different colors for Figure 4, but it can hardly be distinguished. Figure 4 is commended to be updated.

3.      Table 1 is not completed displayed in the manuscript.

Author Response

Dear Reviewer, we thank you for your comments that have allowed us to really improve our work. We hope you will now find it suitable for publication in the JCM.

  1. CT and EOS-imaging provide 3D information of bones. The authors are recommended to more clearly discuss/summarize (such as a table which compares the difference) how that 3D information may improve the accuracy of the parameters obtained by standard radiographs, and how those improvements may be helpful in terms of reconstruction of normal hip biomechanics.

A table has been added (table 2), highlighting each mehods strengths and drawbacks.

  1. Line 139 specified different colors for Figure 4, but it can hardly be distinguished. Figure 4 is commended to be updated.

it has been updated.

  1. Table 1 is not completed displayed in the manuscript.

it has been updated.

Best regards,

RG, corresponding author

Reviewer 2 Report (New Reviewer)

This is an excellent and valuable review of an important topic! Very minor language edits are required. I am particularly impressed with the Figures inserted in the manuscript!

This review manuscript was very well written, it was comprehensive and included many of the key references on the covered topic and the included figures were informative and well chosen.

Author Response

We really thank you for your comments. They come as a reward after hard work to try to deal with this subject, that remains highly variable amongst teams and imaging departments equipments.

Reviewer 3 Report (New Reviewer)

The manuscript can be considered for publication after incorporating minor revisions.

1. Caption of Table 1 will come above it.

2. Introduction part should be improved with latest references pertinent to the topic.

3. Hip deformities Figure 4 should be properly explained.

4. Conclusion part should be rewritten, highlighting the major outcomes of the study.

5. Include the concept of 3d printing in implants.

Author Response

Dear Reviewer, we thank you for your comments, that have allowed us to really improve our work. We hope you fill wind it suitable for publication in the JCM.

  1. Caption of Table 1 will come above it.

Done

  1. Introduction part should be improved with latest references pertinent to the topic.

Done, including references about point 5).

  1. Hip deformities Figure 4 should be properly explained.

Done in the legend.

  1. Conclusion part should be rewritten, highlighting the major outcomes of the study.

Done.

  1. Include the concept of 3d printing in implants.

Done , especially in the CT section.

This manuscript is a resubmission of an earlier submission. The following is a list of the peer review reports and author responses from that submission.

Round 1

Reviewer 1 Report

The paper summarizes the imaging approaches for templating in hip arthroplasty. This is an interesting topic since new technologies enter the field and have the potential to improve patient outcomes.

The paper is well written, in some passages are not connected to the prior passage (eg. line 57).

All abbreviations should be listed in the abbreviation list (e.g. AFD CDA)

Does the Figure caption fit to the template?

Table 2 is to small and not readable.

Indexing of the headlines correct ?(d. Perspectives, i. CT-scan)

Line 176 spaces missing between text and references?

Line 178-179 Sentence sounds weird and should be rewritten.

For coming up with a suitable conclusion it would be needed to include numbers. You talk about dose but do not mention a comparable dose number for each modality. Same counts for costs, time, availability and most important outcome. It is nice to state that a new imaging modality allows more accurate planning. But the decisions for investing in a new technology or not should be based on improvements in outcome, repeatability, costs, workflow....

Please add a section that covers these relevant factors. Only this allows your final conclusion that is worth to be called a conclusion.

Author Response

Dear Reviewer and Journal of Clinical Medicine editorial board,

Thank you for the opportunity to improve our work. In accordance with the reviewers comments the manuscript has been extensively modified, particularly concerning numeric data in order to compare the different imaging modalities studied.

A point-by-point rebuttal to the reviewer’s comments is provided below. All modifications to the text have been tracked.

 I believe that the manuscript has been considerably improved by the modifications requested and I hope that you’ll find it suitable for publication in the Journal of Clinical Medicine in its present version.

I remain at your disposal for any further requirements.

Sincerely,

Leading author

Comments for the Authors:

The paper summarizes the imaging approaches for templating in hip arthroplasty. This is an interesting topic since new technologies enter the field and have the potential to improve patient outcomes.

The paper is well written, in some passages are not connected to the prior passage (eg. line 57).

This passage has been moved to the radiographs section.

All abbreviations should be listed in the abbreviation list (e.g. AFD CDA)

Done

Does the Figure caption fit to the template?

Yes.

Table 2 is to small and not readable.

It has been updated.

Indexing of the headlines correct ?(d. Perspectives, i. CT-scan)

Yes, it is proposed by Microsoft word.

Line 176 spaces missing between text and references?

Done

Line 178-179 Sentence sounds weird and should be rewritten.

 Done

For coming up with a suitable conclusion it would be needed to include numbers. You talk about dose but do not mention a comparable dose number for each modality. Same counts for costs, time, availability and most important outcome. It is nice to state that a new imaging modality allows more accurate planning. But the decisions for investing in a new technology or not should be based on improvements in outcome, repeatability, costs, workflow....

Please add a section that covers these relevant factors. Only this allows your final conclusion that is worth to be called a conclusion.

Numbered data concerning each technique have been added in each section, and allows a better comparison between those methods, as asked.

Reviewer 2 Report

The proposed manuscript is a bit difficult to define. Probably it would be suit to classify it as a narrative review of preoperative templating in hip arthroplasty.  It lacks the necessary details to be a useful description of templating, i.e. a document residents might use as a guide. It lacks any scientific question and the review is not detailed and argumented enough to be a useful review.

In my opinion, the authors should first define the aim of this document. Then, the manuscript should be rewritten accordingly, to focus on the point of interest.

Instead of providing an incomplete list of indications for hip arthroplasty in the Introduction, the authors might better argument what makes hip arthroplasty successful. This may then lead properly to the main body of the text.

Abbreviations should be defined in the text before first use.

Acetabular offset may also be defined from the median line. Easier particularly after total hip arthroplasty and when the cup is in protrusion.

The acetabulum or the cup have a (ante-)version, as both are structures with an opening. A femur has no opening. The proper term for description of the orientation of the femoral neck is (ante-)torsion. Even if this is not applied currently in mostly North American publications.

Regarding CT measurement of the femoral torsion, it is usually easier not to superpose the pictures, but to have them side-by-side. The superposition was originally done due to software limitations of the early generations of CTs. As soon as an open angle measurement tool is available, avoiding superposition makes the pictures more reliable to read.

Referring to the study from Lewinnek et al. (Ref. 17) definitely is inadequate to recommend cup orientation. This study is methodologically somewhere between poor and wrong. Nowadays it may only be used for journal clubs, to illustrate methodological errors. Other references should be used.

Fig. 5 should be corrected. The placement of the lesser trochanter is very creative/artistic but not anatomic.

Author Response

Dear Reviewer and Journal of Clinical Medicine editorial board,

Thank you for the opportunity to improve our work. In accordance with the reviewers comments the manuscript has been extensively modified. Particularly, technical aspects and review references have been reinforced.

A point-by-point rebuttal to the reviewer’s comments is provided below. All modifications to the text have been tracked.

 I believe that the manuscript has been considerably improved by the modifications requested and I hope that you’ll find it suitable for publication in the Journal of Clinical Medicine in its present version.

I remain at you disposal for any further requirements.

Sincerely,

Leading author

Comments for the Authors:

The proposed manuscript is a bit difficult to define. Probably it would be suit to classify it as a narrative review of preoperative templating in hip arthroplasty.  It lacks the necessary details to be a useful description of templating, i.e. a document residents might use as a guide. It lacks any scientific question and the review is not detailed and argumented enough to be a useful review.

As asked, we added technical details and aspects of each imaging modalities to make it clearer. Also, we postulated in the introduction section that we think radiographs remain fundamental to date and cannot yet be replaced by the other imaging modalities, which should be considered as complementary. We added bibliographic references with details and arguments to the review, which in our opinion improved it.

In my opinion, the authors should first define the aim of this document. Then, the manuscript should be rewritten accordingly, to focus on the point of interest.

We described the strengths and limitations of each imaging modality to answer the aforementioned question. The paper has been extensively modified.

Instead of providing an incomplete list of indications for hip arthroplasty in the Introduction, the authors might better argument what makes hip arthroplasty successful. This may then lead properly to the main body of the text.

Done.

Abbreviations should be defined in the text before first use.

 Done.

Acetabular offset may also be defined from the median line. Easier particularly after total hip arthroplasty and when the cup is in protrusion.

Done.

The acetabulum or the cup have a (ante-)version, as both are structures with an opening. A femur has no opening. The proper term for description of the orientation of the femoral neck is (ante-)torsion. Even if this is not applied currently in mostly North American publications.

The term version has been replaced by the term torsion.

Regarding CT measurement of the femoral torsion, it is usually easier not to superpose the pictures, but to have them side-by-side. The superposition was originally done due to software limitations of the early generations of CTs. As soon as an open angle measurement tool is available, avoiding superposition makes the pictures more reliable to read.

The figure has been modified.

Referring to the study from Lewinnek et al. (Ref. 17) definitely is inadequate to recommend cup orientation. This study is methodologically somewhere between poor and wrong. Nowadays it may only be used for journal clubs, to illustrate methodological errors. Other references should be used.

This section has been updated, with clarification about the recent advances in acetabular cup placement.

Fig. 5 should be corrected. The placement of the lesser trochanter is very creative/artistic but not anatomic.

This figure has been updated.

Round 2

Reviewer 1 Report

Thank you for improving the manuscript.

Table 2.  is hard to read, the text seems to be not aligned in the rows?

Line 296 "Practically speaking, without software, a multiplanar reconstruction can be constructed." This sounds misleading. A CT uses software to create a volume dataset out of 2D radiation slices. Do you mean multiplanar planning? and a planning software?

Author Response

Dear Reviewer and Journal of Clinical Medicine editorial board,

Thank you for the opportunity of improving our manuscript (Round 2).

A point-by-point rebuttal to the reviewer’s comments is provided below. All modifications to the text have been tracked.

I believe that the manuscript has been considerably improved by the modifications requested and I hope that you’ll find it suitable for publication in the Journal of Clinical Medicine in its present version.

I remain at your disposal for any further requirements.

Sincerely,

Leading author

Table 2.  is hard to read, the text seems to be not aligned in the rows?

We have added the grid, which makes it easier to read.

Line 296 "Practically speaking, without software, a multiplanar reconstruction can be constructed." This sounds misleading. A CT uses software to create a volume dataset out of 2D radiation slices. Do you mean multiplanar planning? and a planning software?

Thank you for pointing out this point. We have made it more precise.

Reviewer 2 Report

The authors resubmitted a revised version of their manuscript, a narrative review of radiographic modalities available for review. 

I still struggle understanding the aim of this manuscript. It is not a practical guide to templating hip arthroplasty. Radiologic modalities available are reviewed, but no conclusion is provided in which case which modality should be chosen. Of course everything can be done, technically. The clinical need would be to determine appropriate use. The authors fail to define this. The manuscript mostly relies on references describing templating, without going into enough details or justifying certain choices in positioning the implants. Basics in templating are not applied or discussed correctly or with enough detail. There also is no question justifying a study.

The sentence line 66 would need rephrasing. The second part is unclear.

Line 79 is an affirmation by the authors that would need references. The literature by far is not conclusive that the cup should be medialized and the femoral offset increased in compensation. The acetabular offset should not be increasing, due to the risk of lever-out of the cup. While that is well-known, literature probably is inexistent, except some case reports. The sagittal plane has been totally neglected by the authors. Whereas anteroposterior malpositions of the cup are an issue in patients with residual complaints after hip arthroplasty.

Measuring the femoral offset is possible only in the plane of the femoral neck. Thus, it may usually be only estimated on conventional radiographs, particularly when any hip joint pathology is present, as this usually limits internal rotation. The affirmation that this may be measured as described is too much of an approximation.

It remains unclear why the acetabular floor distance should be measured at all. Particularly as it cannot be determined anymore once hip arthroplasty has been performed. The classification proposed in Fig. 3 does not define a range and is as such not applicable.

Femoral stem torsion is just one aspect of restoring hip anatomy. Short stems may be placed in flexion, without any torsion, and still restore a proper position of the centre of rotation. The authors entirely missed this aspect.

The authors recommendation to rely on standing radiographs is questionable. That may be sufficient for most ambulatory patients. But is not adequate for most emergency situations and patients with advanced disease. Also, the variability of standing radiographs is much greater than for supine radiographs. There again, the authors failed to perform in-depth analysis.

The recommendation to accept leg length discrepancies up to 10 mm simply is wrong. Leg length or offset discrepancies above 5 mm are already associated with gait alterations (Renkawitz et al, Gait & Posture 2016 to cite just one reference). The top of the greater trochanter may only be used as a reliable reference if the femur is projected correctly. Otherwise, the centre of the lesser trochanter should be used.

The ranges of Cortical Index values provided by the authors in Fig. 5 would need proper references. These values are not provided in the original publication by Dorr et al. On the other hand, the Dorr classification also relies on the axial view of the hip, not solely on the anteroposterior view. The authors failed to recognized this. The Dorr classification only considers the form of the cortical bone. The quality of the cancellous bone may be even more important. The Singh classification may be valuable for this.

Some abbreviations still need to be defined before use. One good example would be EOS.

This manuscript fails to provide the necessary detail of the state of the art of templating in hip arthroplasty, it fails at providing a critical review of the literature of the important aspects, and it does not provide any new scientific information. 

Author Response

Dear Reviewer and Journal of Clinical Medicine editorial board,

Thank you for the opportunity of improving our manuscript (Round 2). It has been extensively modified and implemented with your recommendations.

A point-by-point rebuttal to the reviewer’s comments is provided below. All modifications to the text have been tracked.

I believe that the manuscript has been considerably improved by the modifications requested and I hope that you’ll find it suitable for publication in the Journal of Clinical Medicine in its present version.

I remain at your disposal for any further requirements.

Sincerely,

Leading author

The authors resubmitted a revised version of their manuscript, a narrative review of radiographic modalities available for review. 

I still struggle understanding the aim of this manuscript. It is not a practical guide to templating hip arthroplasty. Radiologic modalities available are reviewed, but no conclusion is provided in which case which modality should be chosen. Of course everything can be done, technically. The clinical need would be to determine appropriate use. The authors fail to define this. The manuscript mostly relies on references describing templating, without going into enough details or justifying certain choices in positioning the implants. Basics in templating are not applied or discussed correctly or with enough detail. There also is no question justifying a study.

We have added our opinion about indications of CT and we thought we had mentioned the questionable points about CT indications in the limitations section, as no clear recommendation are now available. Precisions have been added.

EOS indications in our opinion are also mentioned, with their advantages and limitations which do not allow us to propose a systematic approach, as EOS utility and results are not yet known for the post-operative follow-up. CT has been investigated in situations where EOS is not reliable, described in the limitations section.

In our opinion and according to the actual literature, we are only able to provide each technique advantages and limitations, to headline the clinical need to perform studies comparing each technique according to clinical scenarios. A direct comparison between EOS and CT would also be of great use in the post operative setting. We do totally agree with you that further scientific data are necessary, but not yet available to our knowledge.

The sentence line 66 would need rephrasing. The second part is unclear.

It has been modified.

Line 79 is an affirmation by the authors that would need references. The literature by far is not conclusive that the cup should be medialized and the femoral offset increased in compensation. The acetabular offset should not be increasing, due to the risk of lever-out of the cup. While that is well-known, literature probably is inexistent, except some case reports. The sagittal plane has been totally neglected by the authors. Whereas anteroposterior malpositions of the cup are an issue in patients with residual complaints after hip arthroplasty.

Reference has been implemented. Acetabular positioning has also been clarified. Sagittal plane references have been implemented.

Measuring the femoral offset is possible only in the plane of the femoral neck. Thus, it may usually be only estimated on conventional radiographs, particularly when any hip joint pathology is present, as this usually limits internal rotation. The affirmation that this may be measured as described is too much of an approximation.

This point has been addressed, as CT-scan measurement is more accurate.

It remains unclear why the acetabular floor distance should be measured at all. Particularly as it cannot be determined anymore once hip arthroplasty has been performed. The classification proposed in Fig. 3 does not define a range and is as such not applicable.

It was proposed by Kase et al. but was removed as it does not get justification in their paper. The cut-off has been added in figure 3.

Femoral stem torsion is just one aspect of restoring hip anatomy. Short stems may be placed in flexion, without any torsion, and still restore a proper position of the centre of rotation. The authors entirely missed this aspect.

This issue has been addressed in the section 2)a) with literature and illustrations.

The authors recommendation to rely on standing radiographs is questionable. That may be sufficient for most ambulatory patients. But is not adequate for most emergency situations and patients with advanced disease. Also, the variability of standing radiographs is much greater than for supine radiographs. There again, the authors failed to perform in-depth analysis.

This point has been treated in the radiograph section 3)a). 

The recommendation to accept leg length discrepancies up to 10 mm simply is wrong. Leg length or offset discrepancies above 5 mm are already associated with gait alterations (Renkawitz et al, Gait & Posture 2016 to cite just one reference). The top of the greater trochanter may only be used as a reliable reference if the femur is projected correctly. Otherwise, the centre of the lesser trochanter should be used.

 References have been added and the value has been modified.

The ranges of Cortical Index values provided by the authors in Fig. 5 would need proper references. These values are not provided in the original publication by Dorr et al. On the other hand, the Dorr classification also relies on the axial view of the hip, not solely on the anteroposterior view. The authors failed to recognized this. The Dorr classification only considers the form of the cortical bone. The quality of the cancellous bone may be even more important. The Singh classification may be valuable for this.

The references are cited in the text (45-47). In the original publication, the cortical index is well shown on the AP view, so we thought it to be a reliable figure. The AP and profile view indexes are correlated according to the original report. The Singh classification has been added to the text.

Some abbreviations still need to be defined before use. One good example would be EOS.

EOS is a commercial denomination; we could not find its significance. The symbol “registered” has been added.

This manuscript fails to provide the necessary detail of the state of the art of templating in hip arthroplasty, it fails at providing a critical review of the literature of the important aspects, and it does not provide any new scientific information. 

We truly think that based on your review, we clearly improved our manuscript and addressed the mentioned issues. We also added technical and practical data. On the other hand, these new imaging techniques require comparison during follow-up, to determine if they can replace standard radiographs.